# Personal Health Record for Personalizing Research and Care Trajectories: A Proof of Concept Pilot with Diet in Inflammatory Bowel Diseases

**DOI:** 10.3390/jpm13040601

**Published:** 2023-03-29

**Authors:** Reinder Broekstra, Marjo J. E. Campmans-Kuijpers, Gerard Dijkstra, Adelita V. Ranchor, Elisabeth W. H. M. Eijdems

**Affiliations:** 1University Medical Center Groningen, Department of Health Sciences, Section Health Psychology, University of Groningen, 9712 CP Groningen, The Netherlands; 2University Medical Center Groningen, Department of Gastroenterology and Hepatology, University of Groningen, 9712 CP Groningen, The Netherlands; 3University Medical Center Groningen, University of Groningen, 9712 CP Groningen, The Netherlands

**Keywords:** personal health record, personalized medicine, precision medicine, return of findings, prevention, research participation

## Abstract

Combinations of health-related research data and clinical data generated, e.g., from wearables, can increasingly provide new insights about a person’s health. Combining these data in a personal health record (PHR), which is managed by citizens themselves, can enhance research and enable both personalized care and prevention. We piloted a hybrid PHR using it for scientific research and the concomitant return of individual findings for clinical information and prevention purposes. The obtained information on the quality of daily dietary intake allowed researchers to further investigate the association between diet and inflammatory bowel diseases (IBDs). Additionally, the feedback enabled participants to adjust their food intake to improve the quality and prevent nutritional deficiency, thereby increasing their health. Our results showed that a PHR including a Research Connection can be successfully used for both purposes but requires a good embedding in both research and healthcare processes with the cooperation of healthcare professionals and researchers. Addressing these challenges is key in the pursuit of delivering personalized medicine and building learning health systems with PHRs.

## 1. Introduction

Rapid advances in information technology have enabled the generation of vast amounts of digital data, on a scale not previously imaginable. Combinations of health-related research data and clinical data generated from, e.g., wearables, can increasingly provide new insights about a person’s health. Combining these data in a personal health record (PHR), which is managed by citizens themselves, can enhance research and enable both personalized care and prevention [1,2]. Despite their projected high adoption rates [3], knowledge about the implications of PHRs is limited. This study piloted the use of a PHR aiming to support both scientific research and care and investigated its implications.

### Background of Personal Health Records

PHRs are internet-based platforms that allow citizens to centrally store and access their own health-related data and insights, which can be exchanged with others, such as healthcare professionals and researchers [4,5,6]. Though the exponential growth of digital data repositories is evident, the centralization of data on health status in a PHR allows an improved use of data and the derived insights in different settings and trajectories. For example, individuals’ health data collected via wearables or questionnaires might be shared with both a health professional improving a clinical care trajectory and with a researcher improving a scientific research trajectory. In the same vein, individuals’ results from the research trajectory can be used to support a clinical trajectory.

Several key characteristics of PHRs can be identified according to the literature. First, individual citizens maintain and control their own record up to a certain point. This distinguishes PHRs from electronic health records and electronic medical records, as these are primarily organized and controlled by healthcare providers [2]. Second, the PHR is designed for data exchange between different systems of healthcare providers or research institutes [4]. Third, PHRs allow citizens to receive and exchange their individual health-related data upon their or researchers´ request or in clinical or research trajectories, which increases information on their individual health status [2]. In contrast, current record systems rarely or never combine data on an individual level from clinical and research settings.

The circular flow of data/information and the increase in knowledge in a PHR, as depicted in Figure 1, supports the prevention of care and clinical treatment tailored on an individual level [1,4]. On the one hand, it can empower citizens with health insights, especially improving the self-management of patients by adding automatic analyses to health-related data and providing tailored feedback. A study from Spil and Klein in 2015 showed that healthcare professionals experienced an improvement in health self-management and patient empowerment if a PHR was used [2]. Other benefits of PHRs mentioned by scholars are, for example, an increase in both the quality and efficiency of care with more individually tailored care, higher patient satisfaction, and healthcare cost reduction [1,4].

These benefits can apply to the research setting as well, since the potential of PHRs to use data and insights for multiple purposes creates a hybrid setting including both research and healthcare. This can enhance research opportunities for scientists by making more health-related data potentially accessible. Better use of data can increase the efficiency and quality of research and can reduce research costs [7]. Moreover, PHRs allow the return of individual findings from research without violating privacy, which can improve research participation and retainment via stronger engagement. Multiple studies found that receiving these personal benefits are an important and stimulating factor in the decision to participate [8,9]. 

The trend of working towards personalized medicine and prevention is seen in the literature and by stakeholders in Europe and the Netherlands [10,11]. Nevertheless, scholars heavily debate the ethical, legal, and societal issues of such information technology used for both research and healthcare purpose or for personalization on an individual level, for example, the use of polygenic risk scores [10,11,12,13]. Few studies have yet investigated how a PHR could and should be built and used for healthcare and scientific research according to stakeholders, especially citizens or patients’ experiences and their attitudes towards a hybrid PHR. Two recent Estonian studies confirmed the potential of hybrid EHRs and PHRs with successful genotype and cascade screening among biobank participants for familial hypercholesterolemia [14] and early detection of hereditary breast and ovarian cancer risk [15]. However, the perspective of the participants and patients on the implications was not further investigated during these studies.

We aimed to close this knowledge gap, so we piloted the build and use of a hybrid PHR to understand patient participants’ views on the main implications of using this technology and the concomitant return of individual research results. Therefore, this study investigated the implications of a hybrid PHR used for scientific data collection for a Dutch cohort and biobank for patients with inflammatory bowel disease (IBD), with the return of individually tailored dietary intake advice. In particular, we focused our study on participants’ first experiences of a pilot with a hybrid PHR.

## 2. Materials and Methods

### 2.1. Data Collection Procedure of PHR-IBD Pilot

#### 2.1.1. 1000IBD

We chose to pilot the hybrid PHR among participants of the 1000IBD cohort, which was initiated as the 1000IBD project (https://1000ibd.org, accessed on 1 July 2022) to prospectively follow more than 1000 patients with IBD from the northern provinces of the Netherlands. For these chronic patients, researchers collected, among other things, information on dietary and environmental factors, drug responses, and adverse drug events [16]. Since evidence-based guidelines are lacking for patients with IBD, they usually experiment with foods themselves to alleviate symptoms, which often leads to nutritional deficiencies [17]. As dietary information is known to be an important factor of the disease, for the 1000IBD project, information on the dietary intake was collected by a specially developed questionnaire for IBD patients called GINQ-FFQ [18]. We transformed this paper-based questionnaire into an online version with the research data capture system REDCap, which is the default research software in the UMCG.

#### 2.1.2. Return of Individual Findings: Diet Intake Score with Explanation

Using the hybrid PHR, we obtained dietary information with the GINQ-FFQ, assessing food intakes over the previous month as a proxy for habitual dietary intake. The NEVO table version 2013, which is the Dutch food composition table, was used to calculate the individual mean composition of food items in grams/day and the average total daily dietary intake [19]. In our study, we compared participants’ daily dietary intake with Dutch healthy dietary recommendations [20] and gave them feedback both on their intakes in terms of amounts of nutrients consumed and how they could potentially enhance intakes of specific nutrients by using more or less of specific food groups. We built an algorithm to automatically generate tailored feedback for participants with a software package in the R language and using the standardized FHIR data exchange protocols [21]. This generated advice provided participants with insight into the quantity of intake (sufficient, almost sufficient, not sufficient) and how to adapt their intake using slider graphics and text (see Appendix A). This questionnaire and concomitant feedback not only detects nutritional deficiencies and confirms a good dietary intake but also allows patients to improve their deficiencies and follow a healthy diet. To complement the findings, the filled-out GINQ-FF questionnaire as a source of data was returned as well.

#### 2.1.3. Personal Health Record and a Research Connection as Missing Link

In our study, we used a PHR, called Mai+Life, and a Research Connection, called Mai+Science, under development by the company Phaitality. The Research Connection collaborates with PHRs handling the request from participants to the research setting and vice versa, for example, to exchange data or return results. This tool thus enables researchers to design and execute a scientific study with participants that use a PHR, while integrating specific research information technology systems.

Current PHRs lack integration with the functionality of research systems, such as authentication and authorization services for researchers, institutional research registries, or research data capture systems. For example, it is not possible with current PHRs to select potential research participants, to send invitations to participate in studies, to execute studies, or to send individual findings from studies. The Mai+Science tool allows one, as shown in Figure 2, to match data within a PHR with data requests and links with authentication and authorization services for researchers, institutional research registries, and research data capture systems. Figure 3 shows a screenshot of its dashboard for researchers.

First, only the scientific researchers from the University Medical Center Groningen (UMCG) had access to Mai+Science despite its aim to provide multiple centers with access to the tool. Only UMCG-certified scientific researchers from the pilot study could use the Research Connection and approach PHRs with requests to protect the data of participants. One of the available technologies in the Netherlands to authenticate and authorize these researchers is Surfconext, which is a federated authentication and authorization technology to ensure secure and user-friendly access to cloud services of different providers. Scientific researchers log in with their institutional account and have secured access to the service of the Research Connection available to them through a single sign-on.

Second, to follow the ethical and legal control regimes for scientific research, it is important to authenticate and authorize the study itself using a PHR. Therefore, Mai+Science needed connections with research registries of institutes integrated in scientific administrative procedures on an institutional level. This ensured that the research of the activities, characteristics, and status of each study, for example, on the status of ethical approval, was controlled and effective. The UMCG research register uses dedicated software called Utopia. The Research Connection is therefore built so that it could verify information on the registration of a specific study and could execute authentication and authorization of study activities, for example, the obtained ethical approval from an institutional review board for conducting a study, and unblocking study activities for registered members involved in the study.

Third, to conduct research and create new knowledge or information in a PHR from data, it is key to link research data capture systems and virtual workspaces of institutes with a Research Connection. Only in these systems can insights be generated and stored in a safe and reliable manner, which becomes especially important in the light of the return of individual findings. In the UMCG, as in many other medical research institutes, REDCap is being used as it is specifically geared to support online and offline data capture for research studies and operations.

For this pilot study, we limited automatic integration of the Research Connection for the research data capture system, since this study focused on the return of individual findings from scientific research in a PHR and the experiences of participants. Therefore, the researchers involved needed to manually link the PHR pseudonym with our study pseudonym and the cohort pseudonym in the Research Connection. This bypassed current issues on pseudonymization and depseudonymization. The researchers could upload the individual results of participants based on the study pseudonym in the Research Connection, thus returning an individual finding from scientific research.

#### 2.1.4. In-Depth Interview with Participants: Understanding the Implications of a Hybrid PHR

To understand the experiences and views of participants on using a PHR and receiving individual findings, we conducted an in-depth interview with all participants. We applied a narrative interview approach with a tailored topic guide that was partly derived from the DIPex methodology that allows for a discussion, clarification, and verification of unanticipated themes [22]. We distillated five relevant themes based on the current state of knowledge in the literature and the explorative character of our study. The themes in our topic guide were: general views on PHR; Mai+Life PHR; general view on return of findings; tailored diet advice and questionnaire; future tailored advice (see Appendix A). This approach can be considered in line with the adaptive theory for qualitative research allowing for the influence of theory on research [23].

#### 2.1.5. Recruitment of Participants and Sampling

A total of 22 participants of the 1000IBD cohort were selected and invited by their doctor face to face, followed up with an invitation email. We selected based on IBD and an earlier expressed willingness to be approached for research. From the invited persons, 11 1000IBD patient participants (50%) participated in our PHR pilot study. Two individuals refused participation for health reasons, and one person refused due to a low self-efficacy in digitalization. Eight individuals did not respond to the invitation email.

After registering an account in the PHR Mai+Life, participants consented explicitly to be available for scientific studies. This allowed their specific PHR to link with our Research Connection. Subsequently, we invited them in their PHR to participate in the pilot study and fill in the GINQ-FFQ. After completing the questionnaire, they received their individual feedback of their filled-in questionnaire in a PDF together with short dietary intake advice to alleviate possible nutritional deficiencies. After feedback on their individual findings, all participants were invited for an interview of 30–60 min about their experience and for their comments. Nine participants consented to the interview, one person did not respond, and one person refused due to limitations in time availability. Table 1 shows the background characteristics of our sample. During our sampling, saturation occurred which was confirmed in both the data collection and data analysis process by not hearing new information in the last interviews or finding new codes. We followed the interpretivist tradition, accepting phronetic or context-dependent knowledge [24].

#### 2.1.6. Data Analysis of In-Depth Interview

We analyzed every phrase in each of the nine interview transcripts within the context of the entire interview, and, where appropriate, a code pertaining to its content was generated or assigned. Codes could be applied multiple times within each transcript, and phrases could comprise multiple codes. Codes with related content were clustered within groups that were subsequently categorized in themes. An initial coding protocol containing a description of the codes, groups, and themes was developed based on a close reading of three transcripts. This coding protocol was evaluated through an iterative process, whereby two or three researchers cross-checked analyses of each of the transcripts independently. Once a consensus regarding the content of the codes, groups, and themes had been reached, the resulting coding protocol was used for the remaining transcripts. This procedure ensured agreement among the researchers regarding the coding. In total, three researchers coded the transcripts. Subsequently, two researchers coded all of the remaining transcripts.

Transcripts were primarily analyzed using the computer-assisted qualitative data analysis package Atlas TI, version 22, Berlin, to retrace and evaluate quotes along with their codes, groups, and themes [25].

## 3. Results

We distinguished four themes relating to the patient participants’ views on the main implications of using a PHR for the concomitant return of individual research results aiming to improve prevention and treatment. The first two themes pertained to the PHR as a new technology in the current data and information flows for patient participants. The first theme was ‘Promising yet complex’ and the second theme was ‘requiring optimization but can act as a double-edged sword’. The last two themes touched on the patient participants’ views relating to the return of individual findings as an incentive for research participation and improvement of personalized care. The third theme was ‘motivating but ambiguous utilization’ and the fourth was ‘trustworthy yet not a replacement for consults’. Table 2 shows part of our coding protocol with example codes of each clustered group of codes which resulted in the mentioned themes.

### 3.1. Personal Health Record as a New Platform

#### 3.1.1. Promising Yet Complex

Most participants perceived the concept of a PHR as a promising system. As one participant explained:


*“Well actually, you can finally put all (your data) in one (system), just for yourself at least. Otherwise I have to look at one application and another portal. Well, I just want to know something … now I have perhaps one system, so one bookmark, one application, which allows me to access all that information.”*
(P5)

In the same vein, some participants reported that a PHR felt more personal compared to portals or other systems, due to the combination of relevant personal information flows. Although the record did not contain much information yet, several imagined it could have much more relevant information on their health.

Nevertheless, its advantages were difficult to grasp for others. Most participants understood, after some deliberation, that a PHR allowed users to exchange health data with different institutes and contexts but found it difficult to imagine examples and to see it as innovative. They often compared the PHR with current ‘patient portals’ of hospitals, for example, the UMCG’s electronic medical record system called myUMCG, based on Epic software. Additionally, participants made a comparison with telemedicine applications, e.g., their IBD application called myIBDCoach [26]. Several participants wondered how the PHR could be distinguished from these systems and if it would not be ‘just’ another system or application. They feared that many similar systems would co-exist, which is not user-friendly for patients, especially concerning login procedures. One participant reported:


*“What I don’t find very inspiring myself is that, as a patient at the UMCG, I now have myUMCG, then I have a separate system for the IBD department called myIBDCoach, which I sometimes have to use to answer questions or look at things. I feel like this was just another one of those things, in addition to the other two, so again with another name and another link and yet another set of passwords.”*
(P2)

#### 3.1.2. PHR Requiring Optimization but Can Act as Double-Edged Sword

Participants considered the specific PHR, Mai+Science, used in this study as a platform which was “easy” to use and felt “familiar”, though should be optimized. Most participants mentioned specifically the registration and login procedures in the study being somewhat cumbersome and a barrier to use. Participant 3 responded vividly:


*“Then I have to log on again, click through, do as I say, look for my password or look in the mail for the right code, and put it in the right place, or then your password expires again, you have to think up a new one. Pff…”*
(P3)

Nevertheless, participants reported that these challenges in registration and use were to be expected due to participation in a pilot and were a more general issue referring to the difficulty of registering accounts with password requirements. All of the participants appreciated that the issues were solved with the help of the support from the research and technology team of the study.

A few participants mentioned that in general digitalization within both research and healthcare was a good development and made things “easier” and “quicker”, especially considering the COVID-19 pandemic. In contrast, another participant perceived this digitalization as somewhat negative, since the systems were less helpful in practice and came at the cost of a live conversation within the clinical setting. The participant explained:


*“And then the two of us are sitting with the specialist and the first minute of the conversation is always spent grumbling about the new, what’s it called, the new, you know, system that the doctor has to work in. Yes, exactly, HP22. Because, what’s it called, the doctor sits staring at his screen endlessly, while it would actually be nice if he could look at me. Because there are a lot of things that need to be ticked off and then ticked off again later. So in the consulting room, we’re actually wrestling with a system that someone has devised, which is supposed to help, but doesn’t really.”*
(P2)

### 3.2. Return of Individual Findings as Incentive for Research Participation and Improvement of Personalized Care

#### 3.2.1. Motivating but Ambiguous Utilization

Participants reported that the promise of the return of individual findings was one of the motivators to participate. Though all participants expressed that without return they would have participated as well, they emphasized that it enhanced their motivation to participate. Most participants explained that they had not previously experienced a return of individual findings from research.

All participants mentioned that they were “curious” to receive the individual result of the diet intake and advice, which was “tailored” and “personal”. They considered diet to be relevant knowledge for patients with chronic inflammatory bowel disease, since they believed in an association between food and this disease based on their personal experience and knowledge. Some participants explained that diet was of particular interest due to this relation, in particular for combating symptoms of their disease. Though all results were received several weeks before by participants, some of them only investigated the individual results just before or during the interview of this study. Despite this postponed reading, several of the participants were very enthusiastic, with one of them explaining:


*“I remember being diagnosed with Crohn’s disease 20 years ago and as a layman and still a young girl I remember asking ‘what about food’. And then they said, ‘well just look at everything that you can handle or what you react well to, it doesn’t matter at all.”*
(P1)

Most participants reported that the results were easy to understand, since the texts were clear and the visualizations were helpful. The diet intake advice itself did not raise questions of understanding. Multiple participants explicitly mentioned that the filled-out questionnaire was convenient for understanding the advice, since it allowed them to somewhat reconstruct the advice based on their input. Their articulations found further support in our study, as the support team did not receive any questions regarding the meaning of the diet intake advice.

#### 3.2.2. Trustworthy Yet Not a Replacement for Consults

The results were considered trustworthy by all participants, even though they could not explain how the intake advice was derived from the results. The reasons mentioned for their trust were, for example, “trust in the recruiting doctor and employees of the Medical Center”, “they had a professional look”, and “the results made sense and substantiated”. When asked about their ideas about future individual findings, some participants had a clear picture of what these could look like. They believed that the tailored results would provide insight into how to improve their health behavior, for example, constructing a full healthy diet or movement plan.

Additionally, they thought that it could benefit their clinical treatment trajectory, and they already wanted to share the result with various treating specialists. In contrast, others found it more difficult to imagine the future but answered that the PHR would be a personal archive containing information they could share with researchers and especially healthcare providers after their consent. A participant explained:


*“I do see the possibilities, to, how do you say that, that you as a patient have the idea okay, I am filling this in now (for research), but that is part of the other things that I fill in (for clinical purposes), that my doctor and the nurse eventually see everything.”*
(P2)

Nevertheless, participants preferred to have some form of online or offline counseling or Frequently Asked Questions added when receiving results. Participant 4 said:


*“And then I’m on one of those joint Facebook pages and we all have a bit of the same questions. Well, we’re all a bit stuck on those (questions). Look, if you can find that (answer) in such a portal or ask: well, what about it? That you then, just for your own, become smarter and to deal with certain things.”*
(P4)

### 3.3. Challenges of Utilization of Research Data for Clinical Use

While the individual results returned were clearly understood, some participants mentioned that they were in conflict with previous diet advice from a nutritionist. In addition, a couple of participants wondered if the advice was tailored enough and what the implications were exactly, taking into account their personal context. For example, one participant explained that having a stoma limits the uptake of certain nutrient groups and would inevitably influence the diet intake possibilities. After some deliberation in the interview, none of the participants was conclusive on whom to ask questions regarding the individual result (dietary intake advice). Some thought that the research team would be the right addressee for questions about the diet intake advice but would lack resources for translation such as time, relevant clinical information, and knowledge of the clinical implications. In contrast, others believed that the treating clinician would be the right addressee, yet they would also lack sufficient resources (time, relevant research information, and knowledge for clinical translation). When asked how to resolve this finding of the research, most participants said to discuss the topic and results with their treating doctor. They hoped that they would get extra help and a referral to a nutrition specialist when possible. Despite the challenges, most participants were convinced that this pilot showed the path to a future where research and care are better connected, as one participant explained:


*“My body has a whole history, you can give as much advice as you like, but it won’t work, it won’t achieve what it should. Now I have to deal with that overlap between research and care again, you have to have everything together.”*
(P6)

## 4. Discussion

In this study, we show that PHR technology can be used to improve diet quality in patients with IBD. It demonstrates that a hybrid PHR system can be used for scientific research and the concomitant return of individual findings for clinical information and prevention purposes. The information on the quality of daily dietary intake obtained allowed researchers to further investigate the association between diet and IBD. Additionally, the feedback enables participants to adjust their food intake to prevent nutritional deficiency, thereby improving their health. Our results showed that a PHR including the Research Connection can be successfully used for both purposes, but this innovation requires a good embedding in both research and healthcare processes. Our study underlines that technological innovation requires organizational innovation as well, particularly considering the European Commission proposals to create a European Health Data Space fostering data use and innovation.

Participants saw the hybrid PHR as a promising technology providing insight into health data and more prevention options, but its value was somewhat complex to understand. This is not surprising as there is a rapid digitalization in the healthcare setting using various records, portals, and applications [27,28]. The field of health data is a Pandora’s box for many citizens. For example, a Danish study from 2019 showed the striking lack of awareness among European citizens about how health data are being used and by whom, but citizens have in general a positive attitude towards the reuse of data [29]. Other studies further confirm the optimistic views on the technology despite the increasing complexity of their implications [30,31], especially for the uptake of PHRs [3]. Our study indicates that the trustworthiness of the PHR as a data repository and the returned findings were not questioned by participants. The participants mentioned the UMCG having a good reputation and their positive experiences with the medical center and the team responsible for the pilot. This amplifies earlier findings on the importance of experiences and institutional reputation for trust in a data repository [9].

Yet, our results showed that participants also perceived this digitalization as a potentially negative development due to the possibility of information overload and depersonalization in clinical trajectories. These concerns align with previously reported concerns of scholars and citizens on digitalization in clinical trajectories, especially the concerns of bias in data and a lack of responsibility [6,32,33,34]. In relation to building trust in a centralized large-scale data repository or artificial intelligence, this last factor is fundamental, especially in healthcare [9,35,36].

The literature shows a similar divide between citizens in relation to trust in information technology, research, and innovation, [34,37,38]. Our findings confirmed the importance of the centralization of data or connection of records as characteristic in decision making about participation in or acceptance of a PHR. We add to this knowledge that the introduction of a hybrid PHR combined with return of findings options motivates patients to adopt the information technology. However, our results underline the threat of a growing polarization between users and non-users. Previous studies robustly showed that unconditional centralization of data or linkage of records is considered a serious threat to privacy according to citizens [1,6,39,40]. Future studies should investigate in particular the issue of polarization when more institutes connect to a PHR and a Research Connection.

Our findings show how these chronically ill patients positively perceive the potential of PHRs, especially the hybrid PHR. The feedback returned helped individuals to: participate in research; to understand their healthy or unhealthy diet intake; and to understand how to improve it. This elucidates previous findings on the general appreciation of the return of results by citizens, for example, in biobank research [41,42,43]. Additionally, it supports the positive view of patients on their telemedicine application [26]. Though participants believed it could be a good innovation, the use might frustrate instead of support citizens’ healthcare trajectories depending on the conditions of implementation. For some people, the feedback on nutrient intake per food group might be enough to make changes in their dietary intake and prevention care, whereas for others it might trigger them to contact other healthcare providers to increase care on this subject. Yet, none of the study participants knew exactly where to discuss these findings.

Our results therefore underline the challenge of how to return results [44] and the importance of communication plans in utilizing research data for clinical purposes [45]. We showed that participants were positive about the return of results, but it is unclear how to cope with these results, including advice. In contrast to their telemedicine application, they lacked a direct point of contact in their PHR, despite being able to send the research team questions or concerns via email [26].

Hence, it is key to proactively provide a dynamic counseling option when these findings are returned. Otherwise this disruption of clinical care might lead to the feeling of depersonalization instead of support in the patient, particularly due to a lack of division of new responsibilities for the interpretation of the personalized research results and translation to treatment implications. This can be achieved either offline or online depending on the communication preferences, but should be integrated technologically, in the PHR itself, and procedurally, in both the research and the clinical trajectories. Considering the European Commission proposals to create a European Health Data Space, these findings underline the importance of an integrated approach for information technological and organizational healthcare innovation.

Even though we probably could not capture all potential thoughts, concerns, and convictions about a hybrid PHR use, our findings do yield insight into the underlying mechanisms of patients’ initial acceptance to use a PHR in both a clinical and research setting. These mechanisms highlight issues in an early adoption stage which indicates the current feasibility of and important implementation or adaption challenges and opportunities for these type of records. Nevertheless, our study has several limitations which should be taken into account before generalizing its results. We conducted this pilot study using a hybrid PHR which is still under development. The experiences of participants were therefore tweaked to this Dutch pilot setting and might differ in other settings, for example, with less trust in biomedical research [46]. It is known that Dutch patients are more open to adopt the PHR as a technology [39]. Additionally, our pilot returned only simple actionable results in a local setting with few participants. Future studies should focus on the opportunities and challenges for feedback of more complicated results, for example, complicated genomic findings, with a larger sample [42]. Moreover, these studies should use more participants and a regional, national, or even international setting, which requires more standardization, e.g., feedback criteria and text and automation, e.g., automatic pseudonymization and pseudonymization solutions.

## 5. Conclusions

Apart from healthcare processes, PHRs can be used for research purposes. Providing patients tailored feedback from research findings enhances patient participation in research and can improve their health status. Embedding PHRs in both a research and healthcare environment can be challenging and requires the cooperation of healthcare professionals and researchers. Addressing these challenges is key in the pursuit of delivering personalized medicine and building learning health systems with PHRs.

## Figures and Tables

**Figure 1 jpm-13-00601-f001:**
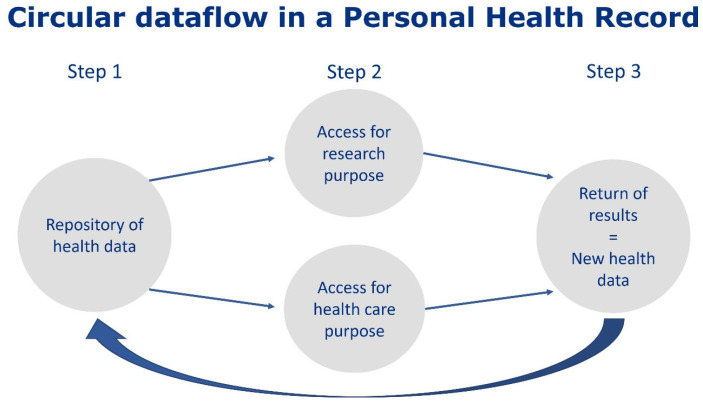
Circular dataflow in a personal health record.

**Figure 2 jpm-13-00601-f002:**
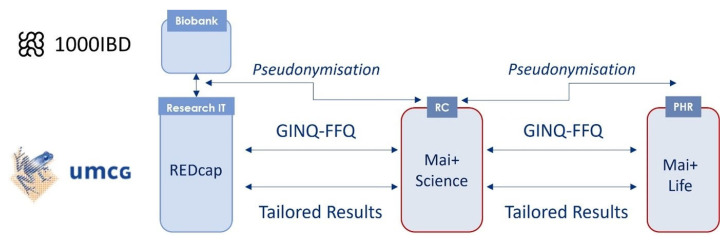
Pilot study design with Research Connection.

**Figure 3 jpm-13-00601-f003:**
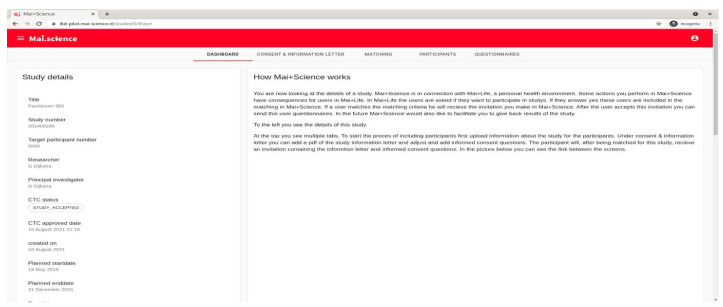
Screenshot of Mai+Science dashboard.

**Table 1 jpm-13-00601-t001:** Characteristics of study participants.

Participant Number	Gender	Age	(Former) Occupational Field	Interviewed
1	Female	35–40	Nursing	Yes
2	Female	50–55	Teaching	Yes
3	Female	35–40	Not expressed	Yes
4	Male	50–55	Telecom	Yes
5	Male	55–60	IT	Yes
6	Female	60–65	Not expressed	Yes
7	Male	45–50	Not expressed	Yes
8	Male	40–45	Horeca	Yes
9	Female	70–75	Not expressed	Yes
10	Male	–	–	No
11	Male	–	–	No

**Table 2 jpm-13-00601-t002:** Coding protocol example.

Themes	Code Groups	Example Codes
**Promising yet complex**	PHR_potential	All data in one place
		Many opportunities
	PHR_another system	Many systems
	PHR_logging in complex	Logging in inconvenient
		Logging in complex
	PHR_for Healthcare provider	Few and different use
		Double effort for patient
**Requiring optimization but can act as double** **-** **edged sword**	Digitalisation_positive	Easy and quick
		Easier
	Digitalisation_negative	Threshold
		Rigid
	Digitalisation_examples	Linking research and care
	Paradox of research & care	Where to go?
	Future_exchange with health care provider	Exchange is convenient
		With consent procedure
	Future_more feedback	Tailored feedback for IBD needs
		Complete feedback
**Motivating but ambiguous utilization**	Results_filled in questionnaire	Double result for patient
		Mainly for health care
	Results_fun, but no action	Something for later
		Own responsibility
	Results_opinions	Advise sometimes tricky
		More specific needs
	Results_follow up actions	Where to go?
**Trustworthy yet not a replacement for consults**	Trustworthiness_research data	No sensitive data
	Trustworthiness_researchers	Feedback from professionals
		Trustworthy via UMCG or acquaintances
	Trustworthiness_results	Return is fun
		Results make sense
	Return_not primary aim	Return not primary aim
		Without return is also fine

## Data Availability

Interview data are available upon request from the corresponding author. Data from the 1000IBD cohort are available via the website: https://www.1000ibd.org/menu/main/home (accessed on 7 January 2021).

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
