# Peer review of "Personal Health Record for Personalizing Research and Care Trajectories: A Proof of Concept Pilot with Diet in Inflammatory Bowel Diseases"

_jpm, 2023, doi:10.3390/jpm13040601_

Round 1

Reviewer 1 Report

Dear Authors,

I see the merit and importance of this work and its contribution to future research and healthcare for patients. However, I did not see any of the Figures nor the Supplementary files as you have stated in your manuscript. The reference list was also missing. 

Nevertheless, based on what I have read and understood, you have tested the feasibility of PHR and acceptability of patients using PHR in a research setting. While I find the methods to be novel, I find the sample size to be too small even for a pilot study; an average sample size for a qualitative study is expected to be 15-20 participants. How certain could you be sure that all the overarching themes are included when you have only 9 participants? Is data saturation achieved? I would also like to see if there is a difference in the responses between men and women. 

I would suggest to include more participants in your pilot study and to resubmit. This will improve the credibility of the pilot and also in restoring my confidence in the results of your study. 

Author Response

Rebuttal to comments jpm-2269870
Reviewer #1 (Comments to the Author):

I see the merit and importance of this work and its contribution to future research and healthcare for patients. However, I did not see any of the Figures nor the Supplementary files as you have stated in your manuscript. The reference list was also missing.

Thank you for your trust in the value of our study. Unfortunately, during the process of structuring the documents and uploading, something went wrong. In the revised version all the documents and element are now complete and should be available in the review platform.

Nevertheless, based on what I have read and understood, you have tested the feasibility of PHR and acceptability of patients using PHR in a research setting. While I find the methods to be novel, I find the sample size to be too small even for a pilot study; an average sample size for a qualitative study is expected to be 15-20 participants. How certain could you be sure that all the overarching themes are included when you have only 9 participants? Is data saturation achieved?

We acknowledge that our sample size is relatively small compared to an average even though our study has an explorative character. We revised our manuscript to amplify this explorative character. Yet, the small sample should not be considered as a problem for our results and conclusions. During our study data saturation occurred confirmed in both the data collection as data analysis process.

Even though we cannot generalize our findings to all potential thoughts, concerns, and convictions on a hybrid PHR use, our findings yield insight in the underlying mechanisms of patients’ initial acceptance to use a PHR in both a clinical and research setting, which indicates current feasibility of an early implementation or adaption of these type of records.

The following papers, which discuss the issue of generalizability, further support our conviction that we reached data saturation in our explorative study:

  • Carminati, L. (2018). Generalizability in Qualitative Research: A Tale of Two Traditions. Qualitative Health Research, 28 (13), 2094-2101. https://doi.org/10.1177/1049732318788379
  • Flyvbjerg, B. (2006). Five Misunderstandings About Case-Study Research, Qualitative Inquiry, 12 (2) , 219-245. https://doi.org/10.1177/1077800405284363

I would also like to see if there is a difference in the responses between men and women.

We did not find significant differences between men and women in their experiences, even after additional analysis. The codes were equally distributed but in general men seem to focus a bit more in their experience on the practical utilization of the return of results while women focused a bit more on the broader implications including the negative. Yet, these differences were only small trends.

I would suggest to include more participants in your pilot study and to resubmit. This will improve the credibility of the pilot and also in restoring my confidence in the results of your study.

Thank you for your suggestion. Though we discussed among the authors if additional data collection increasing our sample size was feasible and needed for our study considering its aim, we concluded that this would not significantly affect our results having the threshold for data saturation reached. We hope our answers to your comments and revision are sufficient to improve the credibility of this study and to restore your confidence in its results.

Reviewer 2 Report

This study is a pilot effort to utilize PHRs for the purpose of realizing truly personalized medicine, and it is meaningful just to make the methodology public. However, in order to publish this article, the following points need to be modified.

p.8-9  It is written that codes were assigned and the codes were clustered, which were subsequently categorized in themes. Then the results of the interviews were categorized in four themes. However those things are not described in Results.  The author should at least clarify what the four themes are and preferably what codes were assigned and how those were clustered and categorized in themes.

p9  It is written that Atlas TI was used to retrace and evaluate quotes, but it is recommended to add more concrete description. The current text does not give details so readers do not see what "evaluating quotes" means.

There are many more, but I cannot give a complete review because of the lack of diagrams and other supporting information. Needs improvement.

Author Response

Reviewer #2 (Comments to the Author):

This study is a pilot effort to utilize PHRs for the purpose of realizing truly personalized medicine, and it is meaningful just to make the methodology public. However, in order to publish this article, the following points need to be modified.

p.8-9  It is written that codes were assigned and the codes were clustered, which were subsequently categorized in themes. Then the results of the interviews were categorized in four themes. However those things are not described in Results.  The author should at least clarify what the four themes are and preferably what codes were assigned and how those were clustered and categorized in themes.

Thank you for your suggestion. We lacked a clear description of the assignment and clustering of codes into themes. In our revision we added this explanation in lines 222 – 232 p.11-12. Additionally we added a table with our coding protocol including some example codes from the code groups within the themes.

p9  It is written that Atlas TI was used to retrace and evaluate quotes, but it is recommended to add more concrete description. The current text does not give details so readers do not see what "evaluating quotes" means.

Thank you for your comment. We hope the current description is better including the topic guide in the supplementary files. We considered to add example quotes to the coding protocol in table 2, but this seemed us to overwhelm the readers. Hence, we left these out in the current revision as some of the example quotes are already in text.

There are many more, but I cannot give a complete review because of the lack of diagrams and other supporting information. Needs improvement.

Thank you for valuable comments given and to come. As answered to the other reviewer as well: Unfortunately, during the process of structuring the documents and uploading, something went wrong. In the revised version all the documents and elements are now complete and should be available in the manuscript text.